# A Coarse-to-fine Framework for The 2021 Kidney and Kidney Tumor Segmentation Challenge

Zhongchen Zhao[1], Huai Chen[2], and Lisheng Wang[1]

Institute of Image Processing and Pattern Recognition, Department of Automation,
Shanghai Jiao Tong University, Shanghai, 200240, P. R. China
13193491346@sjtu.edu.cn

**Abstract.** Kidney cancer is one of the most common malignant tumors in the world. Automatic segmentation of kidney, kidney tumor, and kidney cyst is a promising tool for kidney cancer surgery. In this paper, we use a coarse-to-fine framework which is based on the nnU-Net, and realize accurate and fast segmentation of the kidney and its masses. The average Dice of segmentation predicted by our algorithm is 0.9099, and the Surface Dice is 0.8348.

**Keywords:** Automatic kidney segmentation · Kidney cancer · Coarse-to-fine framework.

## 1 Introduction

Kidney cancer is the 13th most common cancer worldwide, accounting for 2.4% of all cancers, with more than 330,000 new cases diagnosed yearly, and its incidence is still increasing[1]. Due to the wide variety in kidney tumor morphology, it's laborious work for radiologists and surgeons to segment the kidney and its masses manually, Besides, the work relies on assessments that are often subjective and imprecise.

Automatic segmentation of renal tumors and surrounding anatomy is a promising tool for addressing these limitations: Segmentation-based assessments are objective and necessarily well-defined, and automation eliminates all effort save for the click of a button. Expanding on the 2019 Kidney Tumor Segmentation Challenge[2], KiTS2021 aims to accelerate the development of reliable tools to address this need, while also serving as a high-quality benchmark for competing approaches to segmentation methods generally.

## 2 Methods

Semantic segmentation of organs is one of the most common tasks in medical image analysis. There are already many accurate and efficient algorithms for

medical image segmentation tasks, such as nnU-Net [3]. In this paper, we use the nnU-Net as a baseline and adopt the coarse-to-fine strange to segment the kidney, the kidney tumor, and the kidney cyst, as shown Fig. 1. We also propose the surface loss to make the network segment the surface better. To be specific, our algorithm contains three steps:

**Coarse segmentation.** We first use a nnU-net to get the coarse segmentation and crop the kidney region-of-interest (ROI). The kidney masses are always contained in the kidney area. So we can use the kidney ROI, instead of the original full CT image, to get more accurate segmentation results. This step truly matters which can crop the image to a smaller size while still retain the necessary areas.

**Fine kidney segmentation.** We then get the fine predictions of kidney from the cropped kidney ROI by a single classification nnU-net.

**Fine tumor and mass segmentation.** With the kidney ROI and fine kidney segmentation, we segment the kidney tumor and mass by two nnU-Net separately, and combine them with the fine kidney segmentation as the final segmentation.

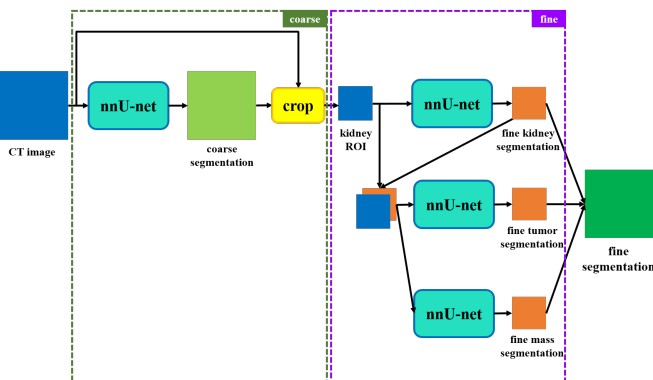

**Fig. 1.** An overview of our coarse-to-fine segmentation algorithm

### 2.1   Training and Validation Data

Our submission made use of the official KiTS2021 training set alone[4]. We divide the provided data into training set and validation set at a ratio of 4:1.

## 2.2 Preprocessing

We follow the way in the nnU-Net to preprocess the training data. The spacing of all official CT images is the same on the x-axis and y-axis, but different on the z-axis. We resample all images to the same target spacing [0.78126 0.78125 0.78125] using third order spline interpolation and then normalize the data. Besides, A variety of data augmentation techniques are applied on the fly during training: rotations, scaling, mirroring, etc.

## 2.3 Proposed Method

### 2.3.1 Kidney Segmentation

After hard mining, we find the accuracy of masses is worse than the accuracy of the kidney. So we decide to segment the kidney mask first and then segment the tumor and cyst from the kidney mask. Specifically, we first use the nnU-Net to get the kidney ROI from the CT image for each case and then use the coarse-to-fine framework to segment the kidney mask from the kidney ROI.

### 2.3.2 Mass Segmentation

With the predicted kidney mask, we use another nnU-Net to segment the masses. By analyzing the provided CT images, we find that the masses are more likely to be on the edge of the kidney. Therefore, we combine the predicted kidney mask containing kidney edge information with the kidney ROI, and feed them to the nnU-Net.

### 2.3.2 Tumor Segmentation

To further improve the tumor segmentation performance, we train another nnU-Net to segment the tumor alone. Similar to the masses Segmentation, we also use the predicted kidney mask and the kidney ROI as the input of the nnU-Net.

### 2.3.3 Surface Loss

Compared to the 2019 KiTS Challenge, the metric this year has one more Surface Dice[5], which is used to quantitatively assess the overlap between the predicted segmentation surface and the real surface. Considering that, we propose the surface loss function, which will penalize the unacceptable regions of the predicted surface. The surface loss function is defined as the sum of distances from all false-positive points and false-negative points in the prediction to the surface of ground truth, as shown below:

$$L_s = \frac{1}{C} \sum_{p_{pred} \in FP \cup FN} \left( \min_{p_{gt} \in S_{gt}} \|p_{pred} - p_{gt}\|_2 \right) \qquad (1)$$

where $S_{gt}$ is the surface of the ground truth, $FP$ and $FN$ are the sets of false-positive points and false-negative points separately, and $C$ is a constant.

During the actual training process, we use the surface loss only when the Dice loss and cross-entropy loss are low enough. In other words, the surface loss is used to fine-tune the surface segmentation results in our work.

### 2.3.4 Postprocess

After all training finished, we remove the isolated blocks which are smaller than 20,000 voxels for the kidney, 200 voxels for the tumor, and 50 voxels for the cyst. We also remove all tumors and cysts outside kidneys and take the rest as the final segmentation results. We don't use the aggregated segmentations from the majority voting strategy.

## 3    Results

### 3.1    Metric

According to the organizers' request, we use two metrics for evaluation, the volumetric Dice coefficient and the Surface Dice. And we use Hierarchical Evaluation Classes (HECs) for the three classes of targets, in which classes that are considered subsets of another class are combined with that class for the purposes of computing a metric for the superset. For KiTS2021, the following HECs will be used:

Kidney and Masses: Kidney + Tumor + Cyst
Kidney Mass: Tumor + Cyst
Tumor: Tumor only

### 3.2    Results on validation set

We summarize the quantitative results in Tab 1. All the results are based on the validation set, which contains 60 cases. The average Dice is 0.9099, and Dice for kidney, kidney masses, kidney tumor are respectively 0.9752, 0.8851, and 0.8693. The average Surface Dice is 0.8348, and Surface Dice for kidney, kidney masses, kidney tumor are respectively 0.9486, 0.7867, and 0.7692. For the tumor segmentation, our algorithm performs significantly better than the baseline. While for the kidney and cyst segmentation, the improvement of our algorithm is very small. This is because we didn't use the cascaded nnU-Net due to its long training time.

**Table 1.** Results of experiments on validation set

| Model | Dice | | | | Surface Dice | | | |
|---|---|---|---|---|---|---|---|---|
| | kidney | masses | tumor | ave | kidney | masses | tumor | ave |
| Baseline | 0.9748 | 0.8793 | 0.8365 | 0.8969 | 0.9499 | 0.7810 | 0.7392 | 0.8234 |
| Ours | 0.9752 | 0.8851 | 0.8693 | **0.9099** | 0.9486 | 0.7867 | 0.7692 | **0.8348** |
| Improvement | +0.04% | +0.58% | +3.28% | **+1.30%** | -0.13% | +0.57% | +3.00% | **+1.14%** |

## 4   Discussion and Conclusion

In this paper, we use a coarse-to-fine framework to segment the kidney, tumor, and cyst from CT images. We use the nnU-Net as a baseline and improve it by using the surface loss and ROI cropping. Experiments show that our improvement truly works. The Dice of the kidney is very high, but the segmentation of kidney masses, especially the cyst, is not accurate enough.

## Acknowledgment

We would like to express our gratitude to the KiTS2021 organizers and the nnU-Net team.

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
