# OpenReview forum: "A Coarse-to-fine Framework for The 2021 Kidney and Kidney Tumor Segmentation Challenge"
_MICCAI.org/2021/Challenge/KiTS — Submitted to KiTS21 Challenge_

### Official Review · Reviewer_EaTY · 2021-08-30

**Rating:** 6

**Review:**

The authors present a coarse-to-fine approach based on nnU-Net which also makes use of a surface loss in order to optimize not only for the volume dice metric but also the surface dice metric. The paper does a good job explaining the approach but it could benefit from the addition of some figures to serve as visual aids for the method and/or some examples of predictions compared with ground truth.

Also, the authors do not mention how they made use of the multiple annotations per instance that were available. Did they use the aggregated segmentations from the majority voting strategy that were made available on github? If so, they should say so to ensure there is no ambiguity. A similar statement should be made about how this was done for validation.

---

### Official Review · Reviewer_s6mv · 2021-08-30

**Rating:** 7

**Review:**

### Overall

- It would be nice if you could expand the abstract by a few sentences. You could mention how this is a submission to the KiTS21 challenge or provide preliminary data on your results, etc.
- I believe the correct acronym for nnUNet is nnU-Net

### Introduction

- Looks good

### Methods

- This section could really benefit from a diagram which visually summarizes your approach. These are especially helpful for coarse-to-fine approaches such as yours
- What spacing did you resample the data to? Which strategy did you use for resampling?

### Results

- It would also be nice to include a figure here which shows an example of your predictions vs the ground truth for one or more cases
- Please also be sure to add your final test set results once they are available

### Discussion and Conclusion

- Looks good

---

### Decision · Program_Chairs · 2021-08-30

**Decision:**

Major Revisions

**Comment:**

Please address the reviewer comments and resubmit